# Atmospheric particulate matter aggravates CNS demyelination through involvement of TLR-4/NF-kB signaling and microglial activation

Bing Han[1†], Xing Li[1†], Ruo-Song Ai[1†], Si-Ying Deng[1], Ze-Qing Ye[1], Xin Deng[1], Wen Ma[1], Shun Xiao[2], Jing-Zhi Wang[2], Li-Mei Wang[3], Chong Xie[4], Yan Zhang[1], Yan Xu[1], Yuan Zhang[1]*

[1]Key Laboratory of Medicinal Resources and Natural Pharmaceutical Chemistry, The Ministry of Education; National Engineering Laboratory for Resource Development of Endangered Crude Drugs in Northwest China; College of Life Sciences, Shaanxi Normal University, Xi'an, China; [2]School of Geography and Tourism, Shaanxi Normal University, Xi'an, China; [3]Department of Neurology, the First Affiliated Hospital of Zhengzhou University, Zhengzhou, China; [4]Department of Neurology, Renji Hospital, Shanghai Jiaotong University School of Medicine, Shanghai, China

**Abstract** Atmospheric Particulate Matter (PM) is one of the leading environmental risk factors for the global burden of disease. Increasing epidemiological studies demonstrated that PM plays a significant role in CNS demyelinating disorders; however, there is no direct testimony of this, and yet the molecular mechanism by which the occurrence remains unclear. Using multiple *in vivo* and *in vitro* strategies, in the present study we demonstrate that PM exposure aggravates neuroinflammation, myelin injury, and dysfunction of movement coordination ability via boosting microglial pro-inflammatory activities, in both the pathological demyelination and physiological myelinogenesis animal models. Indeed, pharmacological disturbance combined with RNA-seq and ChIP-seq suggests that TLR-4/NF-kB signaling mediated a core network of genes that control PM-triggered microglia pathogenicity. In summary, our study defines a novel atmospheric environmental mechanism that mediates PM-aggravated microglia pathogenic activities, and establishes a systematic approach for the investigation of the effects of environmental exposure in neurologic disorders.

*For correspondence:
yuanzhang_bio@126.com

†These authors contributed equally to this work

Competing interest: The authors declare that no competing interests exist.

## Editor's evaluation

The study by Han et al. uses a rodent model of demyelination to investigate the effects of atmospheric particulate matter (PM) on CNS demyelination. The authors provide evidence that PM can promote and exacerbate demyelination which is associated with increased microglial activation and inflammation in the rodent central nervous system. These findings further our understanding of how environmental factors can influence human diseases.

## Introduction

Air pollution has become the prominent environmental risk factor that affects public health (*Caplin et al., 2019*). Among its heterogeneous composition, atmospheric particulate matter (PM) appears to be one of the most harmful components contribute to the pathogenesis of diseases (*Babadjouni et al., 2017*; *Wu et al., 2018*). Although a majority of PM research focuses on the respiratory and

cardiovascular diseases, increasing epidemiological evidence has linked the irreversible adverse roles of continuous PM exposure with the morbidity and mortality of central nervous system (CNS) diseases (*Boda et al., 2020*; *de Prado Bert et al., 2018*; *Younan et al., 2020*).

Studies have confirmed that nanosized PM can physically penetrate the blood-brain barrier and placental barrier, and therefore invade the brain parenchyma of adults and fetuses to directly induce effects in the brain (*Bové et al., 2019*; *Maher et al., 2016*). In 2018, the editorial of The Lancet Neurology reported that air pollution has been associated with increased risk of neurological disorders (*The Lancet Neurology, 2018*). Growing epidemiological reports have indisputably demonstrated that long-time PM exposure affects the development of the nervous system, thus increases the incidence and/or severity of Alzheimer's disease (AD), stroke, brain atrophy, anxiety, and multiple sclerosis (MS), among others (*Babadjouni et al., 2017*; *Boda et al., 2020*; *de Prado Bert et al., 2018*; *Khan et al., 2019*; *Younan et al., 2020*). In addition, due to the specific physiology of fetuses (relatively high ratio of respiration rate to body size and underdeveloped lung), PM exposure during pregnancy or early pregnancy causes long-term and irreversible adverse effects on neurodevelopment of fetus (*Chandrakumar and 't Jong, 2019*; *Sripada, 2017*). However, the pathogenic mechanism of PM leading to different/specific neurological diseases needs to be elucidated in depth.

Myelin sheaths represent key structures for saltatory conduction of nerve impulse and trophic support of axons, and their loss or dysfunction leads to demyelination and impaired neurological function (*Hughes and Appel, 2020*). Clinically, demyelination occurs in a range of human neurological disorders as diverse as MS, optic neuromyelitis, leukodystrophy, spinal cord injury, AD, Parkinson's disease (PD), white matter stroke, as well as schizophrenia (*Goldman and Kuypers, 2015*; *Goldman et al., 2012*; *Malpass, 2012*; *Marin and Carmichael, 2019*; *Mitew et al., 2010*; *Pukos et al., 2019*). The exact etiology and pathogenesis of demyelination diseases are unknown so far, involving many complex factors such as environment exposures, autoimmune response, and genetic susceptibility (*Kumar and Abboud, 2019*). Klocke et al. found that exposure to concentrated fine and ultrafine particles during embryonic development affected oligodendrocyte maturation and brain myelination in adulthood (*Klocke et al., 2017*). Taking MS, a representative inflammatory demyelination disease, as an example, epidemiologic evidence highlights the effect of PM exposure on the risk of incidence, relapse, and deterioration of MS (*Bai et al., 2018*; *Tateo et al., 2019*; *Zhao et al., 2019*), suggesting strong correlations connecting PM exposure to demyelination and remyelination failure. However, the direct evidence of PM action on demyelinating disease is lack, and the cellular/molecular mechanisms leading to this disease process remain unclear.

To investigate this important question, here we develop the complementary pathological and physiological demyelination/myelination models to identify PM that boost microglia-driven neuroinflammation, and define the TLR-4/NF-kB signaling pathways involved in the regulation of microglia pro-inflammatory activities as well as PM-related demyelinating diseases.

## Results

### PM exposure aggravates myelin injury in EAE, a CNS inflammatory demyelination model

Neuroinflammation is one of the major pathogenic factors resulting in CNS injury. To assess the impact of PM aspiration on inflammation-induced demyelination *in vivo*, we first employed experimental autoimmune encephalomyelitis (EAE) model to recapitulate human demyelinating diseases based on immune response. Experimental design and treatment strategies are shown in *Figure 1A*. We found that PM exposure (nasopharyngeal inhalation, 5.0 mg/kg/d) exacerbated EAE progression with an earlier onset compared with the PBS-treated control (*Figure 1B and C*). In PBS group, the average day of onset was day 12 post immunization (p.i.), while in PM-treated group disease onset began at~ day 8 p.i., deteriorated rapidly, and no recovery after reaching the peak (*Figure 1B*). The majority of EAE mice in PBS group displayed moderate signs (limp tail, wadding gait, or paralysis of one limb), while almost all PM-treated mice exhibited severe signs (complete paralysis of both hind limbs or moribund) at the end of observation (*Figure 1C*).

To evaluate the effect of PM on EAE-associated CNS pathology, we isolated thoracic spinal cord sections of EAE mice for histology and immunohistochemistry evaluation. Consistent with the clinical finding, histological analysis revealed significantly enhanced inflammatory and demyelinating foci in

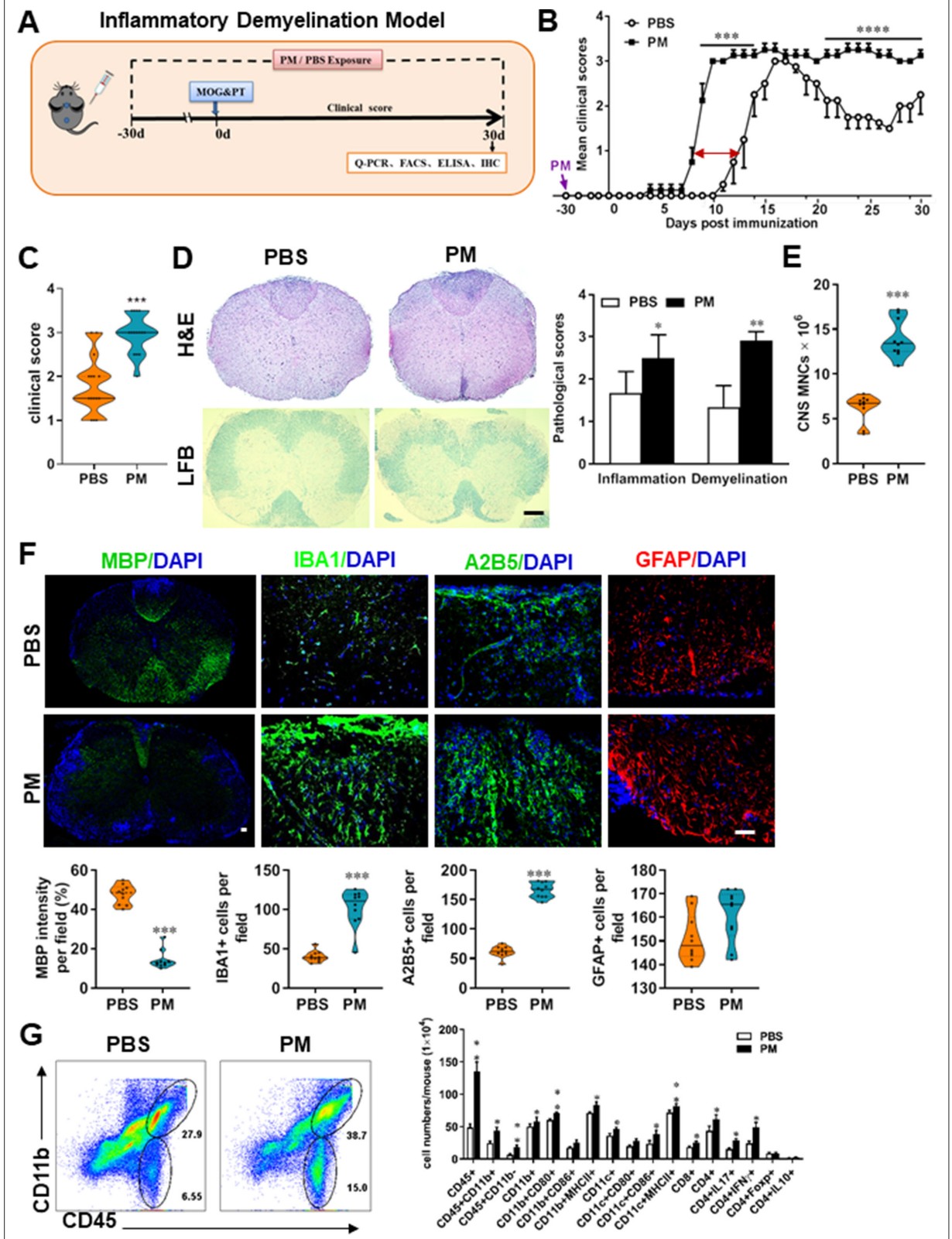

**Figure 1.** PM exposure aggravates myelin injury in an inflammatory demyelination model. (**A**) Schematics of treatment strategies for EAE. Female, 8–10 week-old C57BL/6 mice were immunized with MOG$_{35-55}$ and pre-treated with PBS or PM (nasopharyngeal inhalation, 5.0 mg/kg/d) daily, starting at day –30 (before immunization) until 30 p.i. All mice were sacrificed and their tissues (brain, spinal cord, draining lymph nodes, and spleen) were harvested for Q-PCR, flow cytometry, ELISA, or immunohistochemistry analysis at day 30 p.i. (**B**) EAE development was evaluated daily by two

*Figure 1 continued on next page*

*Figure 1 continued*

researchers blindly, following to a 0–5 scale. (**C**) Distribution of disease status at the end points of experiment (day 30 p.i.). (**D**) Thoracic spinal cord sections were assayed for inflammation by H&E and demyelination by Luxol fast blue (LFB), and CNS pathology was scored on a 0–3 scale. (**E**) Absolute number of CNS mononuclear cells (MNCs) in cell suspension of each mouse (brain and spinal cord) was counted. (**F**) Representative images of spinal cord sections of PBS- and PM-treated EAE mice in the dorsal funiculus. Quantitative analysis of MBP, IBA1, A2B5, and GFAP expression was assessed by using Image-Pro. The measured areas included 8–10 fields and covered virtually all the white matter of the spinal cord. Dorsal column at the thoracic spinal cord is shown as representative images. (**G**) Effects of PM treatment on the various inflammatory cells in the CNS. MNCs from spinal cords and brains were isolated at day 30 p.i., stimulated with $MOG_{35-55}$ (10 µg/mL) for 24 h, and analyzed by flow cytometry. Cells were gated as CD45+CD11b+ (microglia and infiltrating macrophages) and CD45+CD11b- (other infiltrating immune cells), and their subsets were further defined. One representative of three independent experiments is shown. Symbols represent mean ± SD; n = 4–5 mice in each group. *$p < 0.05$; **$p < 0.01$; ***$p < 0.001$; ****$p < 0.0001$, compared to PBS-treated group, two-way ANOVA comparison with Multiple t' tests. Scale bar = 40 µm in D, Scale bar = 10 µm in F.

The online version of this article includes the following figure supplement(s) for figure 1:

**Figure supplement 1.** Effects of PM exposure on different immune cells in periphery.

the white matter of the spinal cord (*Figure 1D*) in PM-treated animals when compared with the PBS control. PM-treated mice also had a remarkably increased number of mononuclear cells (MNCs) in the CNS (*Figure 1E*). The total number of MNCs per mouse in the PM-treated group was 13.9 ± 1.1 × $10^6$, which is ~2.2 fold of that PBS control (6.24 ± 0.8 × $10^6$, $p < 0.001$, *Figure 1E*). We then determined the effect of PM on myelin loss using anti-MBP (myelin marker) staining. As shown in *Figure 1F and a* significant degree of MBP loss (demyelination) had occurred in PM-treated mice, indicating disease progression. Furthermore, significantly accumulation of IBA1+ activated microglia (~2.5 fold in PM-treated EAE mice compared with PBS controls, $p < 0.001$) and A2B5+ OPCs (~2.7 fold in PM-treated EAE mice compared with PBS controls, $p < 0.001$) in the area of demyelination injury were detected in PM-treated mice, while PM aspiration did not significantly impact the expression of the astrocyte marker GFAP (*Figure 1F*). To further evaluate the effects of PM treatment on the various inflammatory cells in CNS, isolated MNCs from CNS were analyzed by flow cytometry. The percentage and absolute number of CD45+ CD11b + cells (microglia and infiltrating macrophages) and other infiltrated immune cells (CD45+ CD11b− cells) were increased obviously in PM group, compared with the PBS control (*Figure 1G*). Increased expression of CD80 and MHCII was also observed in CD11b + cells (microglia/infiltrating macrophages) and CD11c + cells (dendritic cells; DCs) (*Figure 1G*), indicating an enhanced activation of these cells. The total numbers of CD4+ and CD8+ T cells and percentages of Th17 (CD4+ IL17+) and Th1 (CD4+ IFN-γ+) cells were also significantly increased under PM aspiration (all $p < 0.05$; *Figure 1G*). To study the effect of nasopharyngeal inhaled PM on the peripheral immune response, splenocytes of EAE mice were stimulated with $MOG_{35-55}$ peptide and analyzed by flow cytometry. As shown in *Figure 1—figure supplement 1*, splenocytes in PM-treated group exhibited significantly increased numbers and expression of co-stimulatory molecules (e.g. MHC class II, CD80, CD86) of antigen-presenting cells (CD11c+), and higher percentage of IFN-γ+, IL-17+, and TNF-α+ CD4+ T cells. We also examined the effect of PM on Th17 and Th1 cell differentiation *in vitro*, and similar to LPS stimulation, PM treatment significantly increased Th1 cells but had no significant effects on Th17 cells (*Figure 4—figure supplement 1A*). Overall, these results showed that PM exposure possibly aggravates inflammatory demyelination of the CNS.

## PM aspiration exacerbates demyelination and prevents remyelination in a toxin-induced demyelination model

We then investigated the impact of PM aspiration on coprizone-induced demyelination, another classical mammalian animal model that for studying pathological processes associated with demyelinating diseases. Experimental design and treatment paradigms were shown in *Figure 2A*. As shown in *Figure 2B*, body weight of mice in PBS group showed progressive loss during the 4 weeks of cuprizone fed. After cuprizone withdrawn and returning to normal diet for 2 weeks, weight loss was gradual recovery and remyelination was observed in the corpus callosum area of mice in the PBS group (*Figure 2B and C*). However, continuous PM aspiration obviously inhibited the weight gain (*Figure 2B*). PM exposure significantly increased the susceptibility to demyelination injury caused by cuprizone (oral administration for 3 weeks), and prevented CNS spontaneous remyelination in the corpus callosum even after cuprizone withdrawn for 2 weeks, as detected by FluoroMyelin staining (*Figure 2C*). The numbers of CNS resident immune cells, GFAP+ astrocytes and IBA1+ microglia, were

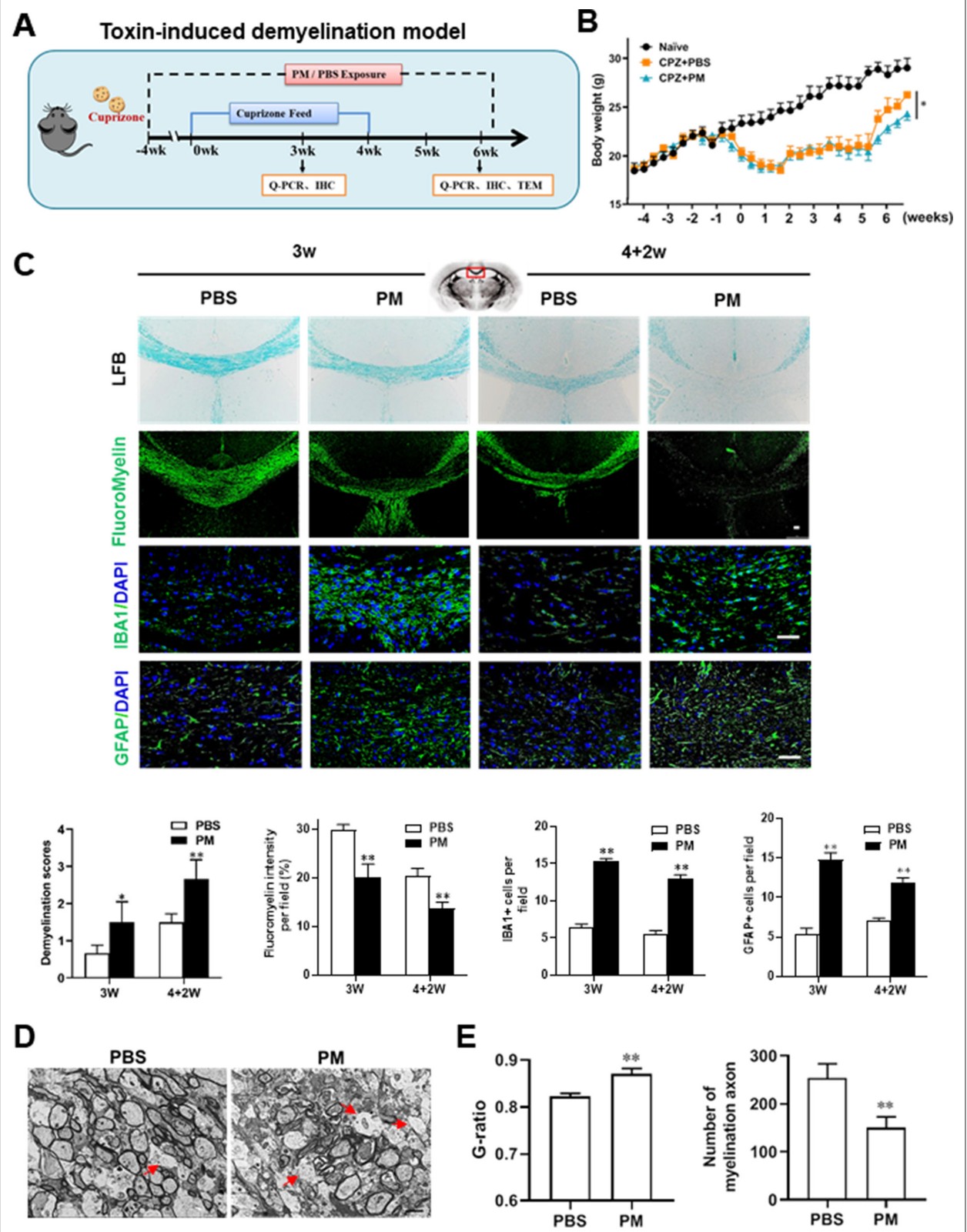

**Figure 2.** PM aspiration exacerbates demyelination and prevents remyelination in a toxin-induced demyelination model. (**A**) Treatment paradigms. Male, 8–10 week-old C57BL/6 mice were pre-treated with PBS or PM (nasopharyngeal inhalation, 5.0 mg/kg/d) daily from week –4 to week 6. Standard rodent diet containing 0.2% copper chelator cuprizone (CPZ), which causes CNS demyelination, were fed for 4 weeks to achieve complete demyelination in the corpus callosum; cuprizone was then withdrawn and mice were again fed normal chow, allowing for spontaneous remyelination to

*Figure 2 continued on next page*

*Figure 2 continued*

occur within the next 2 weeks. (**B**) Body weights of mice from different groups were recorded every 2 days. (**C**) Representative images and quantitative analysis of LFB, FluoroMyelin, and immunohistochemistry (GFAP, IBA1) stains in the body of the corpus callosum of PM- or PBS-treated mice at different timepoints. (**D**) Representative electron microscopy images of the corpus callosum region isolated from PM- or PBS-treated mice at 2 weeks after cuprizone withdrawal. Arrows heads mark demyelinated axons. (**E**) Quantification of the myelinated axons shown in (**D**). The G-ratios (axon diameter divided by entire myelinated fiber diameter) of myelinated fibers and number of myelination axon were assessed by using Image-Pro. One representative of three independent experiments is shown. Symbols represent mean ± SD; n = 5–8 mice in each group. *$p < 0.05$; **$p < 0.01$; ***$p < 0.001$, compared to PBS-treated group, two-way ANOVA comparison with Tukey's multiple comparisons test. Scale bar = 50 μm in C, Scale bar = 2 μm in D.

also significantly increased in the PM-treated group both at demyelination and remyelination process, indicating that PM exposure exacerbated neuroinflammation and demyelination in toxic-induced demyelinating mice, and significantly inhibited myelin repair (*Figure 2C*). Under 4-week-cuprizone treatment and 2-week normal chow, most of the axons in the corpus callosum re-wrapped by thin myelin sheath (spontaneous remyelination) in PBS-treated group, while in PM group only few axons with loosely wrapped myelin were observed as evaluated by ultrastructual electron microscopy (EM; *Figure 2D*). PM aspiration not only significantly reduced the number of myelination axon, but also increased G-ratio of the remyelinated axons, indicating a poor recovery from demyelination (*Figure 2E*). These findings suggest that PM aspiration exacerbates demyelination and prevents remyelination in toxin-induced demyelination model.

## PM exposure during pregnant and postnatal delays myelinogenesis in the developing nervous system

Given that the respirable PM aggravated myelin injury under pathological conditions (immune- and toxicity-induced demyelination), we then investigated whether maternal PM exposure during pregnancy affected the postnatal myelination in the developing nervous system under nonpathological conditions by employing the neonatal mouse myelinogenesis model. We hypothesized that PM aspiration during pregnant and neonatal period would inhibit precocious oligodendrocyte differentiation and myelination; therefore, we chose the postnatal day 14 to examine the extent of myelination in the developing corpus callosum, because the establishment of the myelin sheath completes within the first 3 weeks postnatally in rodents (*Osorio-Querejeta et al., 2017*; *Figure 3A*). As shown in *Figure 3* B, C, 14 days after birth the extent of MBP intensity in the corpus callosum of PBS-treated group were achieved 33.4% ± 8.5% per field, while PM treatment significantly decreased the intensity of MBP staining to 10.1% ± 2.6%, indicative of a 70% reduction in myelination compared with the PBS-treated littermates at this time point ($p < 0.001$). The numbers of IBA1+ microglia and GFAP+ astrocytes were also increased significantly following the PM treatment, indicating the widespread inflammation in the brains of offspring mice and the critical role of micro- and astrogliosis in myelination/demyelination (*Figure 3B and C*). Lower numbers of myelination axons as well as thinner myelin sheathes were found in the PM treatment group than those of the PBS-treated littermates at postnatal Day 14 by EM (*Figure 3D and E*). The expression levels of pro-inflammatory cytokines such as *Tnfa*, *Il1b*, *Il6*, and *Inos* were remarkably increased in PM-treated mice, with significantly inhibited myelination and myelin protein gene expression compared with the PBS control (*Figure 3F*). These pathological changes prompt us to find out if the delayed myelin development induced by PM exposure resulted in any adverse effects in neurological function. Balance and motor coordination were evaluated in a rotating rod, beam walking, and tight rope test. In the rotating rod test at accelerating speed, PM-treated mice exhibited a significant shorter latency to fall off the rod than that of PBS-treated mice (decreased –32.6% in the time of staying on the rotating rod, *Figure 3G*, $p < 0.001$). Disturbances of motor dysfunction in PM-treated mice were also observed in the beam walking and tight rope test. Mice exposed to PM had significantly worse motor performance, as the time taken by mice to traverse a narrow beam or cotton rope increased remarkably compared to PBS-treated control mice (*Figure 3G*, $p < 0.05$). These results clearly indicate that maternal exposure to PM during pregnancy have adverse effect on fetal and neonate mouse, which lead to pathological and structural changes in the myelin development, activation of astrocytes and microglia, thus directly contributes to dysfunction of movement coordination ability of mouse offspring.

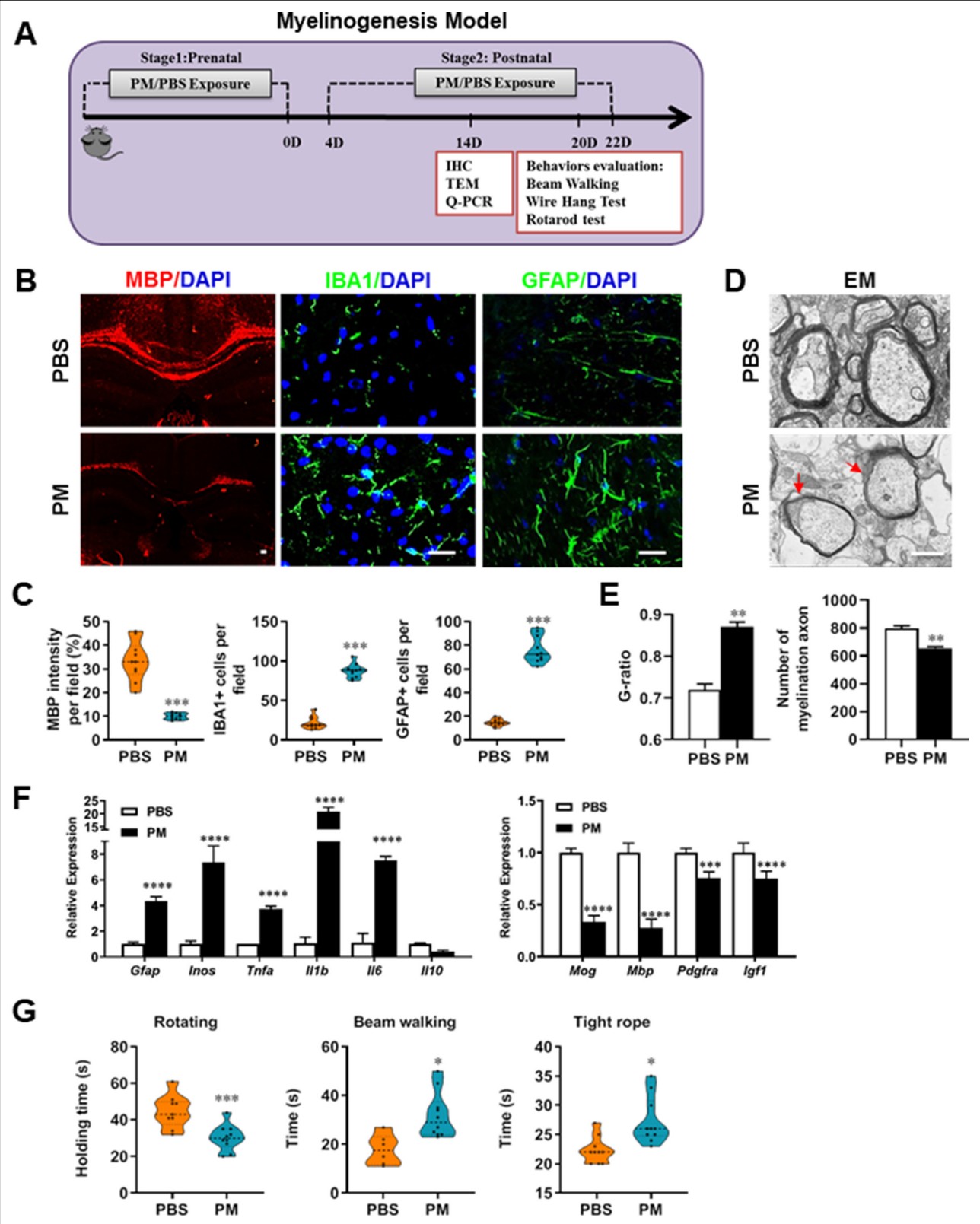

**Figure 3.** PM exposure during pregnant and postnatal delays myelinogenesis in the developing nervous system. (**A**) Schematics of treatment strategies. In order to model a maternal PM exposure, pregnant mice were pre-treated with PBS or PM (nasopharyngeal inhalation, 5.0 mg/kg/d) daily until parturition. Pups from PBS- or PM-treated group with similar weights were subsequently exposed to PBS or PM at postnatal Days 4–21. Brain was harvested for Q-PCR, immunohistochemistry, or TEM analysis at postnatal day 14, and behavioral evaluation was processed at postnatal Days 20–22. (**B**)

*Figure 3 continued on next page*

*Figure 3 continued*

Representative images of myelin content (MBP staining), microgliosis (IBA1 staining), and astrogliosis (GFAP staining) in the body of the corpus callosum of PM- or PBS-treated mice. (**C**) Quantitative analysis of MBP, IBA1, and GFAP expression using Image-Pro. (**D**) Representative electron microscopy images of the corpus callosum region isolated from PM- or PBS-treated mice. Arrows heads mark myelin thinning and dysplasia. (**E**) Quantification of the G-ratios (axon diameter divided by entire myelinated fiber diameter) of myelinated fibers and number of myelination axon by Image-Pro. (**F**) mRNA relative expression of pro-inflammatory cytokines and myelin protein in corpus callosum of PM- or PBS-treated mice was detected by real-time PCR. (**G**) Effect of PM on motor balance and motor coordination were determined by behavioral evaluation (rotating rod, beam walking, and tight rope test). Data were collected from three to five separate mouse litters. Symbols represent mean ± SD; *$p < 0.05$; **$p < 0.01$; ***$p < 0.001$; ****$p < 0.0001$, compared to PBS-treated group, two-way ANOVA comparison with Tukey's multiple comparisons test. Scale bar = 50 μm in B, Scale bar = 1 μm in D.

## PM exposure boosts microglial activation and pathogenicity

Although little is known about the mechanism of PM exacerbating myelin damage, we have observed a significant increase of activated IBA1+ microglia in the demyelination/myelinogenesis animal models (*Figures 1F, G, 2C and 3B* and C). There are similar pathological features in human autopsy study with high levels of air pollution (*Knochelmann et al., 2018*), suggesting that abnormal activation of microglia may be the main cellular event and pathogenic factor in the process of PM promoting neuroinflammation and demyelination. As the resident innate immune cells and sentinels surveying the CNS environment, microglia respond quickly to a vast repertoire of stimuli, including environmental toxins, wound, pathogens, or cellular damage (*Wolf et al., 2017*). The abnormal activation of microglia under PM exposure in the demyelination/myelinogenesis animal models led us to hypothesize that the polarization states may have critical roles in regulating the de/remyelination. To test this, primary microglia were stimulated by exposure to PM, using endotoxin lipopolysaccharide (LPS) as a positive control. It is well accepted that LPS is a classic stimuli to induce microglia into proinflammatory phenotype. Interestingly, incubation of PM with the primary microglia could mimic, even completely replace, the effect of LPS (*Figure 4A–D*). Most of the cells in PBS group were spindle-shaped with slender branches, while under LPS or PM treatment, the cell body became larger and rounder with decreased and thickened protrusions, presenting typical 'amoeba-like' activation morphologies (*Figure 4A*). Similar to LPS stimulation, PM-treated microglia expressed higher levels of MHC II and CD86 (*Figure 4B*). Because upregulation of pro-inflammatory factors or markers is one of the hallmarks of microglial activation, we also examined the polarization by gene expression profiling and enzyme-linked immuno-absorbant assay (ELISA). As shown in *Figure 4C and D*, both PM- and LPS-primed microglia are characterized by expression of pro-inflammatory cytokines and markers like *Tnfa*, *Il6*, *Il1b*, *Inos*, *Ccl2*, *Ptgs2*, and *Stat3*, with remarkably decreased gene expression level of anti-inflammation markers (*Il10*, *Arg1*, and *Il4*), indicating that PM stimulation, similar to LPS, promotes microglia activation and induces cell switch from a resting state to a pro-inflammatory phenotype.

These observations prompted us to explore whether PM exposure of microglia resulted in functional changes as well as disease-promoting activity. To test this, we assessed the responses of microglia-CD4+ T cell co-culture as well as purified primary OPCs to application of microglia-conditioned media (MCM) *in vitro*. As shown in *Figure 4E*, PBS-, LPS-, or PM-activated microglia were co-cultured with purified naïve CD4+ T cells in the context of Th17 differentiation. LPS- and PM-primed microglia Similar to LPS stimulation, PM-treated microglia led to comparable induction of IL-17 and IFN-γ secretion by CD4+ T cells (*Figure 4E*). In addition, OPCs treated with control medium (OPC differentiation medium) or PBS-MCM for 7 d became mature MBP⁺ oligodendrocytes with normal branching of the processes, while the addition of LPS- or PM- MCM prohibited OPC differentiation, evidenced by the significantly decreased numbers and branch score of MBP⁺ cells (*Figure 4F*). These data indicate that PM and LPS may have analogous biological activities and act on the same signal pathway of microglia priming. In addition, we explored the effects of PM on primary astrocytes and OPC *in vitro*. Quantitative PCR showed that the expression of *H2-d1*, *Gbp2*, and *Srgn2* genes, markers of activated astrocyte, was significantly upregulated under LPS treatment compared with the vehical group, but not in the PM-treated group, indicating that PM stimulation cannot promote astrocyte activation like LPS (*Figure 4—figure supplement 1B*). For OPC differentiation, compared with the control group, the number of CNPase⁺ mature OLGs decreased and the number of PDGFRα⁺ OPC increased in the PM treatment significantly (*Figure 4—figure supplement 1C* and D). Fewer branches and poor film formation structures were observed in the PM treatment (*Figure 4—figure supplement 1E*). The inhibitory effect of PM on OPCs differentiation and extension of processes was dose-dependent.

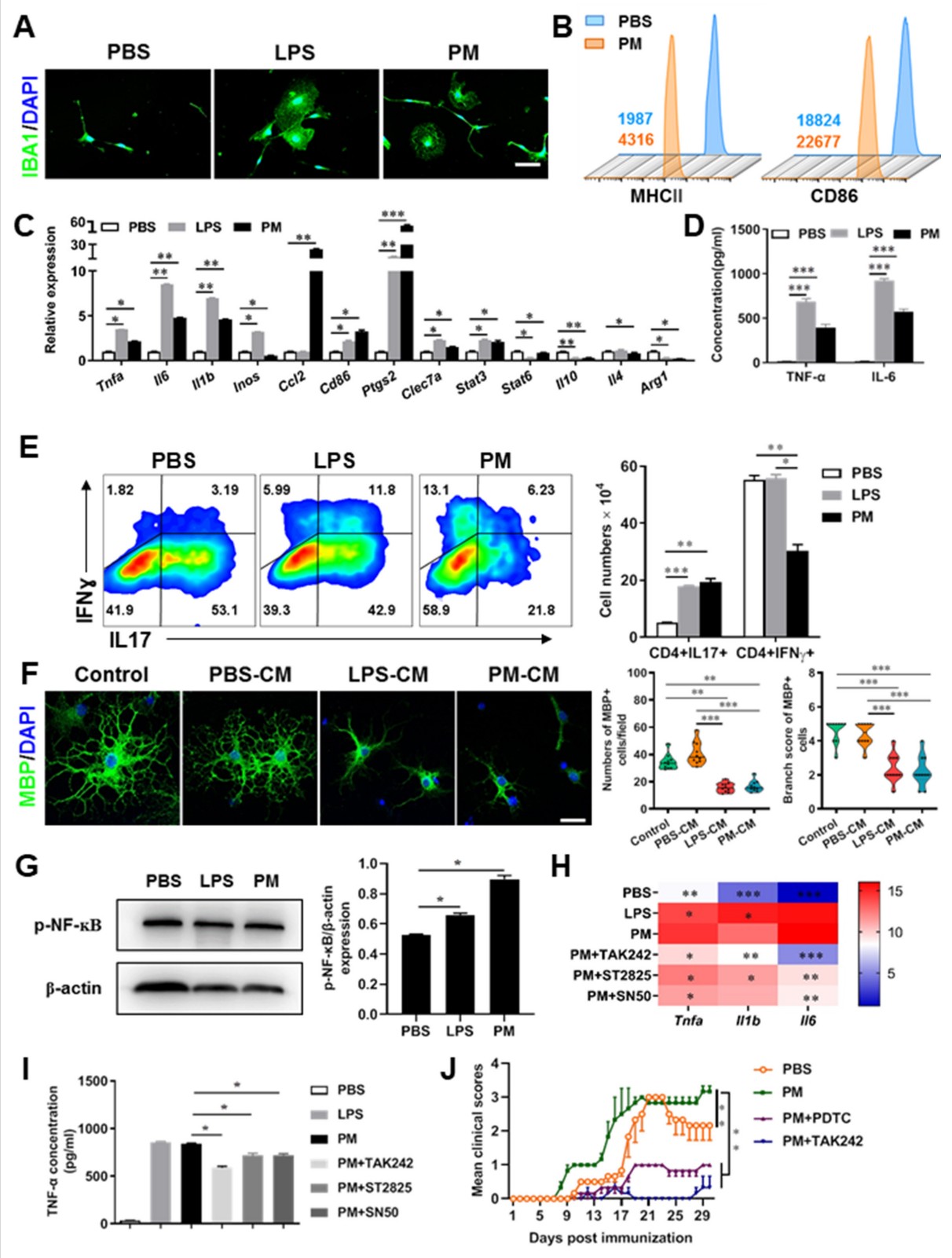

**Figure 4.** PM exposure boosts microglial activation and pathogenicity. (**A**) Morphology of microglia under the treatment of PM or LPS. Primary microglia from newborn C57BL/6 mice were stimulated for 6 h with PBS, LPS (100 ng/mL) or PM (100 μg/mL). Microglia/macrophages were labeled with IBA1 (green) and nuclei stained with DAPI (blue). (**B**) Effects of PM treatment on the expression of MHC II and CD86 of microglia. Primary microglia were stimulated with LPS or PM for 24 h, and analyzed by flow cytometry. (**C**) mRNA relative expression of cytokines and cell surface markers of LPS- or

*Figure 4 continued on next page*

*Figure 4 continued*

PM-treated microglia was detected by real-time PCR. (**D**) ELISA analysis for production of pro-inflammatory cytokine TNF-α and IL-6 in culture medium under PM treatment. (**E**) Naïve CD4+ cells were cocultured with LPS- or PM-activated microglia under Th17-polarizing conditions. The percentage of Th17 and Th1 cells in the CD4 subset were analyzed by intracellular staining of IL-17 and IFN-γ, respectively. (**F**) Responses of primary OPCs to application of microglia-conditioned media (MCM) *in vitro*. OPCs (5000 cells/cm²) were cultured in differentiation medium for 3 days, and half of medium was replaced by culture supernatants of microglia treated with PBS (PBS-MCM), LPS (LPS-MCM), or PM (PM-MCM) for another 4 days. Mature oligodendrocytes were identified by specific markers MBP (green). Quantitative analysis of numbers or branch score of MBP+ mature oligodendrocytes was assessed by using Image-Pro.(**G**) Immunoblot analysis of the phosphorylation of NF-kB (**p–NF–kB**). (**H**) mRNA relative expression of pro-inflammatory cytokines (*Tnfa*, *Il6*, and *Il1b*) of LPS- or PM-stimulated microglia pre-treated with TAK242 (1 µg/mL), ST2825 (10 µM), or SN50 (10 µM) for 4 h. (**I**) ELISA analysis for production of TNF-α in LPS- or PM--stimulated microglia pre-treated with TAK242 (1 µg/mL), ST2825 (10 µM), and SN50 (10 µM) for 4 h. (**J**) EAE development was evaluated daily for PM- or PBS-treated group with or without TAK242 and PDTC administration by two researchers blindly, according to a 0–5 scale. Female, 8–10 week-old C57BL/6 mice were immunized with MOG$_{35-55}$ and pre-treated with PBS or PM (nasopharyngeal inhalation, 5.0 mg/kg/d) daily, starting at day –30 (before immunization) until the end of the experiment. TAK242 (5.0 mg/kg/d) or PDTC (10.0 mg/kg/d) was given by intraperitoneal injection at day 0 p.i. Data were collected from 3 to 5 mice each group. Symbols represent mean ± SD; *$p < 0.05$; **$p < 0.01$; ***$p < 0.001$; two-way ANOVA comparison with Tukey's multiple comparisons test. Scale bar = 100 µm in A, Scale bar = 50 µm in E.

The online version of this article includes the following figure supplement(s) for figure 4:

**Figure supplement 1.** Effects of PM exposure on primary astrocytes, OPCs, and T cell subsets.

TLR-4 (Toll-like receptor-4), as the immunoreceptor of LPS, is a critical regulator of microglia activation. After activation of TLR-4 on the microglia membrane, a cascade of signal transduction may phosphorylate the downstream nuclear transcription factor NF-kB, leading to increased expression of pro-inflammatory cytokines such as *Tnfa*, *Il6*, *Il1b*. As shown in ***Figure 4G–I***, PM treatment, similar to LPS, significantly upregulated the phosphorylation of NF-kB (p-NF-kB) and the expression of pro-inflammatory cytokines (*Tnfa*, *Il6*, and *Il1b*) in cultured microglia. However, the induction was blocked by TAK242, ST2826, and SN50, inhibitors that specifically impede the intracellular activities of TLR4, Myd88 and NF-kB, respectively, in the TLR-4/ NF kB signaling pathway (***Figure 4G and H***). In EAE model, administration of TAK242 and PDTC, pharmacological inhibitors of TLR4 and NF-kB, effectively reversed the deterioration of the disease resulted from PM exposure (***Figure 4J***). Taken together, these findings suggest that TLR-4/NF-kB signaling drives PM-induced microglia activation.

## Mechanism of PM-induced microglia activation via TLR-4/NF-kB signaling axis

Microglia plays important roles in the pathological demyelination, as well as myelination or remyelination under physiological condition. Our cytological data of PM action on microglia identified TLR-4/ NF-kB signaling as a candidate regulator axis of microglia pathogenic activities. To evaluate the role of TLR-4/NF-kB pathway in PM-stimulated microglia, we first proceeded to investigate the PM-specific effects on gene transcription on microglia by using RNA sequencing (RNA-seq, ***Figure 5A–E***). Results shown that 145 genes were upregulated and 57 were downregulated under PM perturbation compared with PBS control (fold change >2; false discovery rate [FDR] < 0.001; ***Figure 5A***). As expected, the upregulated mRNAs in PM-treated group encode a cohort of LPS response genes and inflammatory regulators, including *Mmp2*, *Lif*, *Nod2*, *Nfkb1*, and *Socs3*, consistent with the microglia activation phenotype (***Figure 5A–C***). Similarly, real-time qPCR analysis confirmed a remarkable induction in TLR-4/NF-kB signaling associated genes, such as *Nod2*, *Tnfrsf1b*, *Tnfaip3*, *Fas*, and *Nfkb1* (***Figure 5B***). Gene Ontology (GO) enrichment analysis showed that the upregulated genes in PM-treated group are enriched for those that function in cellular response to LPS and biotic stimulus, positive regulation of defense response, and activation of inflammation-associated pathways, while the downregulated genes are enriched for those that function in negative regulation of cell proliferation and inflammatory response (***Figure 5C and D***). The gene ontology analysis demonstrated that PM-regulated genes in microglia positively associated with Toll-like receptor, NF-kB, TNF signaling pathway, as well as cytokine-cytokine receptor interaction in KEGG pathways (***Figure 5E***).

Some of the effects on NF-kB regulation can be direct, whereas others can reflect indirect events. To better distinguish these possibilities, chromatin immunoprecipitation sequencing (ChIP-seq) was performed to assess the direct transcriptional targets and genome-wide occupancy of NF-kB in PM-stimulated microglia (***Figure 5F–H***). The majority of NF-kB binding peaks were present in intron (31.15%) and intergenic (34.43%) regions and only 21.31% of that located at promoter regions

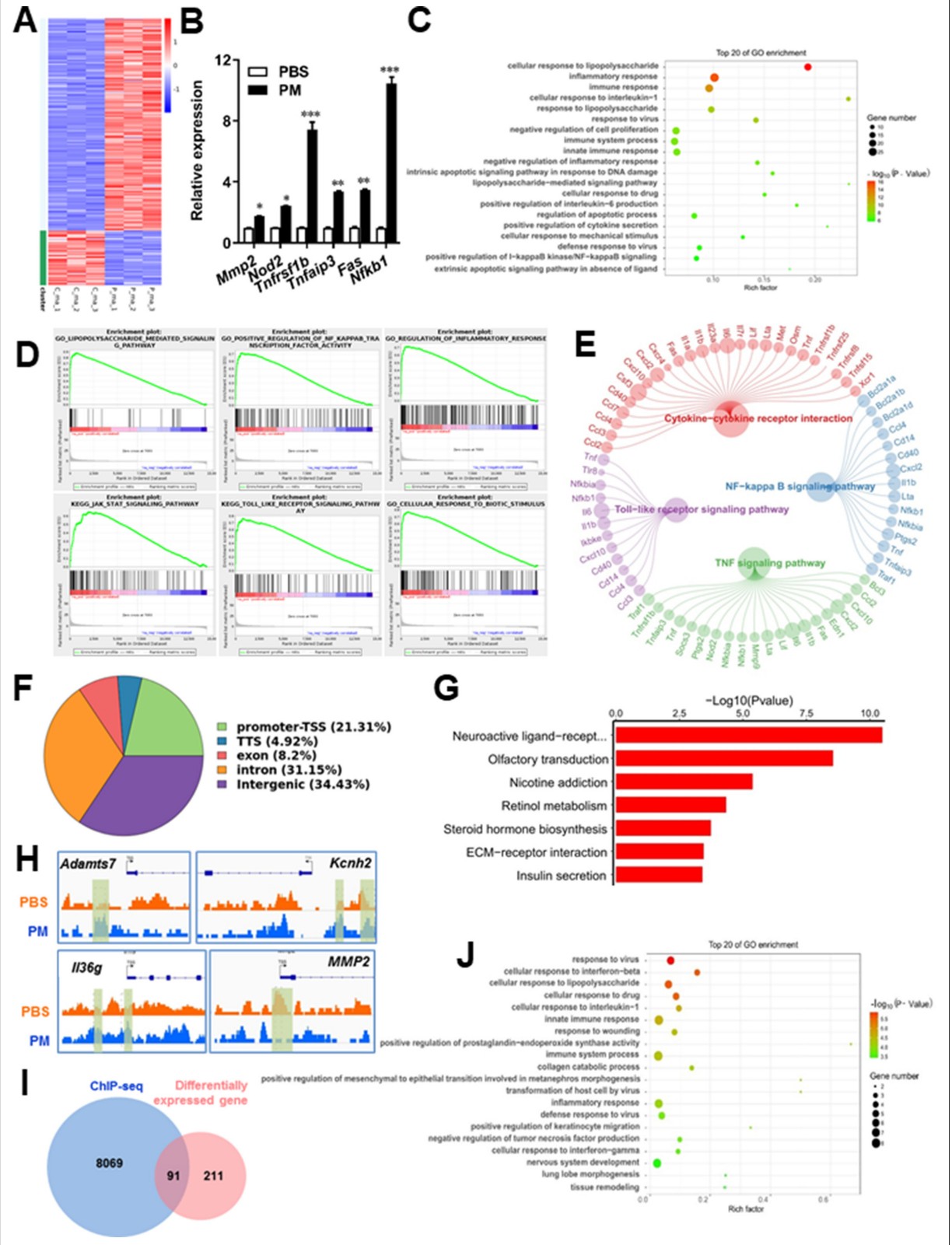

**Figure 5.** PM-induced genomic regulation microglia activities by TLR-4/NF-kB signaling. (**A**) Heatmap displaying the fold changes of genes (rows) in the PM perturbations (columns). Displayed are only genes that were differentially expressed (fold change >2; false discovery rate [FDR] < 0.001) in at least one condition. (**B**) Real-time qPCR analysis of PM-regulated genes in PM and PBS-treated microglia (n = 3). (**C**) The Gene Ontology (GO) analysis of the significantly regulated genes between PM and PBS-treated group. (**D and E**) Gene set enrichment analysis (**D**) and pathway analysis (**E**) of PM

*Figure 5 continued on next page*

Figure 5 continued

and PBS-treated microglia RNA-seq. (**F**) Fractions of ChIP-seq peaks in different regions of the genome. (**G**) GO of NF-kB-targeted genes. (**H**) ChIP-seq showing NF-kB binding at selected gene loci (*Asamts7*, *Kcnh2*, *IL1f9*, and *Mmp2*). (**I**) Venn diagram showing the overlap between NF-kB-bound genes and differentially expressed genes in PM and PBS-treated microglia. (**J**) GO functional categories analysis of NF-kB directly targeted genes.

(*Figure 5F*). GO analysis showed a significant enrichment for genes involved in neuroactive ligand-receptor interaction, suggesting potential NF-kB-targeted gene network in microglia specifically responding to PM stimulation (*Figure 5G*). Among NF-kB-targeted genes were those encoding factors involved in the ADAMTS family (a disintegrin and metalloproteinase with thrombospondin motifs, *Adamts7*), potassium voltage-gated channel family (*Kcnh2*), interleukin one family member (*Il36g*), and matrix metalloproteinase gene family (*Mmp2*) (*Figure 5H*). There were 91 genes overlapped between those affected by PM perturbation and directly bound by NF-kB; among them, 77 genes were upregulated and 14 genes were downregulated (*Figure 5I*). The bound targets were also highly enriched with cellular response to stimuli (virus, cytokines, LPS, drugs, wounding etc.) and immune responses (*Figure 5J*). Together, these data suggest that NF-kB-associated inflammation signaling mediated a core network of genes that control PM-triggered microglia priming.

## Discussion

Demyelination is the common pathological features of several neurological diseases. Although the pathogenic factors are exactly unknown so far, environmental trigger, especially for atmospheric PM, gained increasing attention in neuropathology of demyelination diseases. This is evidenced by the observations that exposure to higher levels of ambient airborne PM was epidemiologically associated with the incidence and development of common demyelination diseases such as MS, optic neuromyelitis, leukodystrophy, and white matter stroke clinically (*Babadjouni et al., 2017*; *Boda et al., 2020*; *de Prado Bert et al., 2018*; *Khan et al., 2019*; *Younan et al., 2020*). However, the mechanisms underlying how inhaled air pollution modulates CNS-resident cells to contribute to the pathogenesis of neurologic diseases is complex and poorly understood. Therefore, to create targeted and effective therapies, mechanism of action (e.g. specific signaling transduction and responsors) that mediates the exacerbating or mitigating clinical symptoms of disease needs to be elaborated.

In the present study, we aim to identify critical cellular and molecular targets that alter demyelination disease after PM exposure, to identify the therapeutic strategies that may be particularly applicable to patients who are exposed to high levels of PM. We demonstrate that PM exposure exacerbates CNS myelin injury, based on three complementary animal models, the immune-induced EAE model, the toxicity-induced demyelination model that under minimal or non-inflammatory micro-environments, as well as the myelinogenesis model during postnatal development. As summarized in *Figure 6*, this is the first comprehensive description of rodent *in vivo* responses to atmospheric PM, under pathological and physiological condition, which shows excessive boost of microglia via the TLR4/NF-kB signaling axis.

We found that PM aspiration obviously increased the expression of proinflammatory factors (e.g. *Il6*, *Il1b* and *Tnfa*), induced activated IBA1+ microglia, decreased MBP intensity in the brain, and impaired normal oligodendrocyte maturation and function. Moreover, maternal PM exposure induced the activation of astrocytes and microglia and subsequent neuroinflammation and myelin dysplasia in the brain of mouse offspring. *In vitro*, PM exposure enhances microglial activation and pathogenicity, whereas PM did not stimulate astrocyte activation. Our results indicate that the effect of PM on glial cells is mainly through microglia rather than astrocytes. Meanwhile, we found that PM exposure also had direct inhibitory effect on OPC differentiation. Consistent with our results, recent studies have reported that air pollutants including PM caused neuroinflammatory responses, promoted demyelination, and caused AD-like pathologies and brain impairment both in adult mouse and offspring (*Calderón-Garcidueñas et al., 2004*; *Chen et al., 2018*; *Ku et al., 2017*; *O'Driscoll et al., 2018*), indicating that the brain, and in particular the glial cells, may be compromised by PM exposure during developmental windows.

Surprisingly, it was found that the types of PMs result in differential outcomes in CNS damage. For example, O'Driscoll et al. reported that chronic dosing of intranasal SRM1649b PM was not sufficient to worsen severity of EAE but did delay onset of EAE, while acute dosing of intranasal SRM1649b

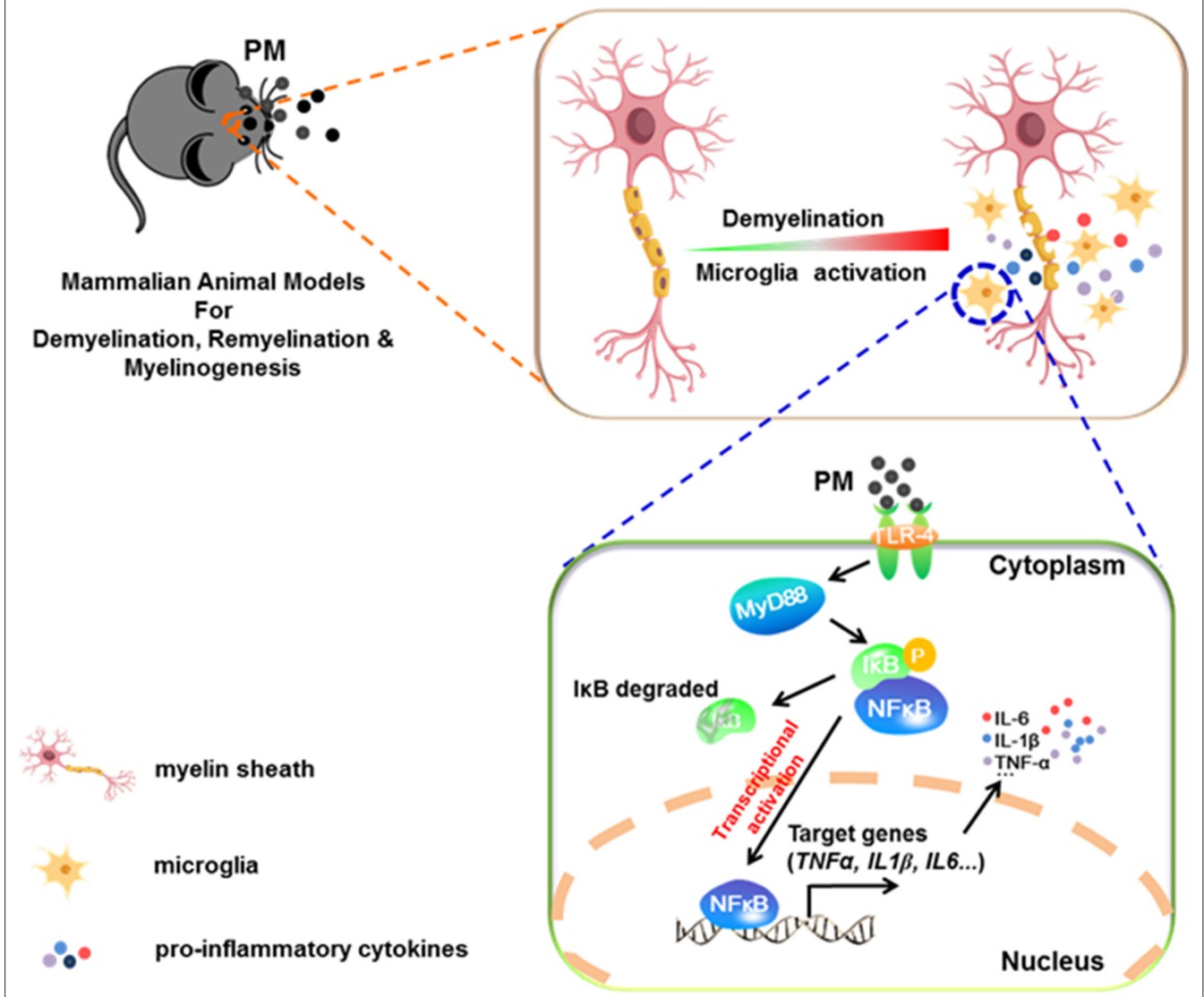

**Figure 6.** Model of PM aggravates CNS demyelination via TLR-4/NF-kB-mediated microglia pathogenic activities. PM exposure definitely exacerbates CNS myelin injury, based on three complementary animal models: the immune-induced EAE model, the toxicity-induced de/remyelination model that under minimal or non-inflammatory microenvironments, and the myelinogenesis model during postnatal development. The cellular basis of this action is associated with the activation of microglial pro-inflammatory activities. Mechanistically, TLR-4/NF-kB signaling mediated a core network of genes that control PM-triggered microglia pathogenicity. Activated microglia, the resident CNS immune cells, respond to PM perturbation directly, release pro-inflammatory factors, and subsequently aggravate neuroinflammation, myelin injury, and dysfunction of movement coordination ability of mice.

PM reduced severity of EAE (*O'Driscoll et al., 2019*). At the same time, they found that two different diesel PM samples enhanced Th17 differentiation and aggravated EAE (*O'Driscoll et al., 2018*). The authors concluded that the active components of PM, not the total mass, are the crucial factor mediating the biological responses.

Our data demonstrated that the cellular mechanisms of PM aggravated the neuroinflammation and demyelination could be mediated by excessive microglial activation by producing neurotoxic pro-inflammatory factors. Microglia are the resident innate immune cells in the CNS, which respond to perturbation caused by environmental stimuli, toxins, trauma or diseases hypersensitive, and performs continuous monitoring (*Yeh and Ikezu, 2019*). A number of clinical and neuropathological studies have shown that priming microglia exhibit a typical pro-inflammatory phenotype, which is one of the key pathogenic factors during aging and in a variety of CNS-diseases including AD, PD, MS, amyotrophic lateral sclerosis, and stroke (*Wolf et al., 2017*). Consistent with our results, recent studies have

shown that PM exposure induced neuroinflammatory including increase inflammatory cytokine secretion *in vivo* and *in vitro* (*Morgan et al., 2011*; *Woodward et al., 2017b*). Our study extends these results in the demyelination lesion specificity of the PM response.

Guided by the gene expression analysis, we defined a role for PM in the activation of TLR4/NF-kB-driven pathogenic activities in microglia. PM enhances pathologic microglia activation in a TLR4/NF-kB-dependent manner leading to worsened demyelination disease in a murine model of EAE. TLR4/NF-kB signal pathway is involved in the regulation of multiple important physiological and pathological processes, such as immunity, inflammation, tumorigenesis, aging, and neurological diseases (*Mitchell et al., 2016*). NF-kB is a ubiquitous DNA-binding transcription factor, which has differential biological effects depending on the extent and duration of activation (*Mitchell et al., 2016*). It is also a typical inducible transcription factor, thus the direct target gene profile and corresponding regulatory networks are variable with different stimuli (e.g. LPS, TNF-α, IL-1β or PM) among different types of cells (e.g. lymphocytes, fibroblasts, epithelial cells or microglia) (*Martin et al., 2020*). In other words, the target genes regulated by NF-kB is stimuli and cell-type specific. Although we found PM incubation with microglia mimic the effect of LPS, which present the typical 'amoeba-like' activation morphologies and high expression of pro-inflammation cytokines and markers like *Tnfa*, *Il6*, *Il1b*, *Inos*, *Ccl2*, *Ptgs2*, and *Stat3*, the target gene profile of PM and LPS is distinct. These findings are consistent with the exacerbation of lung LPS inflammatory responses when combined with PM exposure (*Woodward et al., 2017a*), indicating that the target gene responses of PM were not only due to endotoxin presence in the PM samples, but may also be attributed to the high chemical heterogeneity of PM. TLR4/NF-kB signaling serves as a candidate regulatory axis for PM-induced pathogenic activity in microglia. PM treatment significantly induced TLR4/NF-kB signaling-related genes, such as *Nod2*, *Tnfrsf1b*, *Tnfaip3*, *Fas*, and *NFkb1*. *Nod2* has been demonstrated to trigger microglial activation via the TAK1-NF-kB pathway (*Wang et al., 2020*). *Tnfaip3* could inhibit TNF- and TLR4-induced NF-kB activation (*Vereecke et al., 2009*). Stimulation of group II metabotropic glutamate 2 (mGlu2) in primary microglia induces a neurotoxic microglial phenotype, and *Fas* ligand enhances the neurotoxicity (*Taylor et al., 2005*). However, the relationship between the up-regulation of these genes and the health hazards of PM is still unclear, and the function of *Tnfrsf1b* in microglial activation is unknown. Our further study will focus on the function of PM-indused NF-kB-targeted genes in the process of microglial activation. Subsequently, the alleviating and therapeutic effects of candidate genes on PM-induced CNS myelin injury will be evaluated.

Taken together, our present study confirms that PM inhalation leads to aggravate CNS demyelination, and this action is associated with a previously unrecognized role for TLR4/NF-kB signaling-mediated microglia activation. The results suggest a novel mechanism for PM-produced adverse effects on the nervous system and present a potential intervention target for prevention. Importantly, given the specific nature of PM, e.g., the biologic responses of this complex mixtures is further influenced by the source and constituents, the route of exposure, the particulate matrix within which they reside, the potential different mechanisms and bio-availability of these components, and the genetic differences of the recipients, we believe that the investigation of these differences is necessary to clarify the characteristics of PM exposure and the potential to cause CNS disease. Only then will it be reasonable to propose targeted remediation to stem the tide of demyelination disease that is growing in populations facing air pollution.

## Materials and methods
### Mice
C57BL/6 mice (8–10 weeks of age) were purchased from the Fourth Military University (Xi'an, China). All experimental procedures and protocols of mice were approved by the Committee on the Ethics of Animal Experiments of Shaanxi Normal University (No. ECES-2015–0247) and were carried out in accordance with the approved institutional guidelines and regulations.

### Particulate matter (PM) sample preparation
The PM standard reference materials (SRMs) 1648 a were obtained from the National Institute of Standards and Technology (NIST) (Gaithersburg, MD). Dispersed suspensions of SRM1648a were created by sonication in sterile phosphate buffered saline (PBS) for 15 min in a cooling water bath. SRM1648a

PM was used at 5.0 mg/kg/d PM per dose for *in vivo* experiments or used at 100 µg/mL PM at the highest concentration *in vitro*.

## Animal treatment

Considering the actual population exposure dose and the previous study (*Ku et al., 2017*), we chose the administration dose of 5.0 mg/kg/d PM exposure in mice. According to Ambient air quality standards of China (GB3095-2012), the amount of Grade II PM at 0.15 mg/m$^3$/d is 0.04 mg. It is reported that the respiratory volume of the mice was 90 mL/min and the respiratory volume was ~0.26 m$^3$. Thus, the PM exposure dose in mice is ~2.0 mg/kg/day. In our study, the PM dose used was 5.0 mg/kg/day, which was 2.5-fold higher than that in Grade II PM in China but still in the range of the reported maximum PM levels (*Xia et al., 2020*).

## EAE induction and PM treatment

Female, 8–10 week-old C57BL/6 mice were immunized with MOG$_{35-55}$ and pre-treated with PBS or PM (nasopharyngeal inhalation, 5.0 mg/kg/day) daily, starting at day –30 before immunization until 30 p.i. Mice were immunized at two sites on the back with 200 µg of myelin oligodendrocyte glycoprotein peptide 35–55 (MOG$_{35-55}$) (Genescript, Piscataway, NJ) in 200 µl of emulsion containing 50% complete Freund's adjuvant with 5 mg/ml heat-killed *Mycobacterium tuberculosis* H37Ra (Difco, Lawrence, KS). All mice were intraperitoneally (i.p.) injected with 200 ng pertussis toxin (Sigma-Aldrich, St. Louis, MO) in PBS on days 0 and 2 p.i. Clinical EAE was scored daily in a blind manner, according to a 0–5 scale as described previously (*Yang et al., 2009*): 0, no clinical signs; 0.5, stiff tail; 1, limp tail; 1.5, limp tail and waddling gait; 2, paralysis of one limb; 2.5, paralysis of one limb and weakness of another limb; 3, complete paralysis of both hind limbs; 4, moribund; and 5, death. EAE mice were randomly enrolled in the following treatment groups: (1) Sham-treated PBS control group: EAE mice were exposed intranasally to 20 µL PBS; (2) PM-treated group: EAE mice were exposed to 5.0 mg/kg/d PM in a total of 20 µL PBS per dose. The mice were dosed with 20 µL of PM or PBS treatment three times starting at day –30 days until day 30 after induction. Mice were anesthetized and perfused with 4% PFA; brains and spinal cords were removed for histopathological, immunohistochemistry, electron microscopy, Q-PCR, FACS, and ELISA analyses.

## Cuprizone-induced demyelination and PM treatment

For the cuprizone model, 8-week-old male C57BL/6 mice were fed the standard rodent diet containing 0.2% copper chelator cuprizone (CPZ) for 4 weeks, which causes CNS demyelination. For induced remyelination experiments, mice were fed cuprizone for 4 weeks to achieve complete demyelination of the corpus callosum, after which cuprizone was withdrawn and mice were again fed normal chow, allowing for spontaneous remyelination occurring within the next 2 weeks. PM (5.0 mg/kg/d) or PBS was nasopharyngeal inhaled daily from –4 week to 6 week. Mice were anesthetized and perfused with 4% PFA; brains were removed for histopathological, immunohistochemistry, electron microscopy, and Q-PCR analyses.

## PM exposure of pregnant mice and their offspring

In order to model a maternal PM exposure, pregnant mice were pre-treated with PBS or PM (nasopharyngeal inhalation, 5.0 mg/kg/d) daily until parturition. Pups from PBS- or PM-treated group with similar weights were subsequently exposed to PBS or PM at postnatal Days 4–21. Brain was harvested for Q-PCR, immunohistochemistry, and TEM analysis at postnatal Days 14, and behavioral evaluation was processed at postnatal days 20–22.

## Behavioral experiments

In order to evaluate the motor balance and motor coordination of mice, beam walking test, rota-rod test, and wire hang test were done at the postnatal 19[th], 20[th], and 21[th] day, respectively. Beaming walking test and wire hang test have been done between 8 a.m. to 12 a.m, and rota-rod test has been completed between 2p.m to 6p.m. After each trail, all devices were wiped clean with 75% alcohol to prevent interference with the next trail.

For beam walking test (*Skripuletz et al., 2015*; *Skripuletz et al., 2010*), each mouse was placed on the end of a 100 cm long steel beam with 10 mm wide (placed horizontally 60 cm above a foam

cushion). The one end of beam mounted on a support and the other attached to the rat cage which mice could escape into. Mice received two trials which each trial interval 60 s and the latency to traverse the beam was recorded for each trial (cut-off time 60 s). In the results, the mean score of the two trials is given.

In the rotarod test (*Scoles et al., 2017*), mouse was put to the rotarod with a speed of 5 rpm for 5 min 1 day in advance to adapt to the rotarod. In the formal trial, mice were placed on the rotarod which has an initial speed of 5 rpm and accelerates at an increase of 1 rpm per second. The latency that the mouse stuck before they fell was recorded. Mice received two trials which each trial interval 60 s, and the mean score of the two trials is given in the final results.

For Wire hang test (*Shao et al., 2019*), each mouse was placed on a cotton rope (50 cm long, 2 mm diameter) that connected to two 60 cm high platforms. A foam cushion was placed just below the rope to prevent mice from injuring. Each mouse was put in the middle of the rope and the latency to reach one of the platforms was recorded (cut-off time 60 s). Mice received two trials which each trial interval 60 s, and the mean score of the two trials is given in the final results.

## Histological analysis

Lumbar spinal cords or brains were harvested for pathological assessment. CNS tissues were cut into 7 μm sections, fixed with 4% paraformaldehyde, and stained with hematoxylin and eosin (H&E) for assessment of inflammation, and with Luxol fast blue (LFB) for demyelination. Slides were assessed and scored in a blinded fashion for inflammation (*Yang et al., 2009*): 0, none; 1, a few inflammatory cells; 2, organization of perivascular infiltrates; and 3, abundant perivascular cuffing with extension into the adjacent tissue. For demyelination quantification, total white matter was manually outlined, and area (%) of demyelination was calculated using Image-Pro Plus software.

## Electron microscopy

Mice were deeply anesthetized and perfused with 4% PFA, 1.5% glutaraldehyde and 1 mM $CaCl_2$ in 0.1 M cacodylate buffer. Brain or section of ventral spinal cords was harvested and fixed in the same solution at 4 °C for 24 h. Samples were washed, post-fixed with 1% OsO4 in 0.1 M PBS (pH 7.4) for 2 h at room temperature, and subsequently dehydrated in graded ethanol series. Embedding was performed in TAAB resin. Sections, 1.0 μm thick, were cut, stained in toluidine blue (1%), and examined by light microscopy (E800, Nikon) for general histological assessment. Ultrathin sections (60–80 nm) were cut, viewed and photographed with a HT7700 (Hitachi) transmission electron microscope operated at 120 kV. Images were analyzed in Image-Pro for thickness of myelin sheath and g-ratio.

## Immunofluorescence

For immunohistochemistry, spinal cord or brain tissues were fixed using 4% paraformaldehyde for 1 day and then cryo-protected using a 30% sucrose solution for 3 days. Fixed tissues were embedded in OCT compound (Tissue-Tek, Sakura Finetek, Japan) for frozen sections and then sectioned coronally at 12 μm thickness. Transverse sections of brain and spinal cord were cut, and immunohistochemistry was performed using different Abs following established procedures.

For immunocytochemical staining, microglia medium was fixed with 4% paraformaldehyde for 30 min at room temperature and washed twice with PBS. Cells were permeabilized with 0.3% Triton X-100 (in PBS) for 5 min at room temperature and washed twice with PBS. Sections were incubated with 10% goat serum in PBS for 30–60 min; primary Abs were then added and incubated at 4 °C overnight. Primary Abs were washed out with PBS three times after overnight incubation. Sections were then incubated with species-specific secondary Abs for 60 min at room temperature, followed by washing with PBS three times. Immunofluorescence controls were routinely prepared by omitting primary Abs. Nuclei were stained with DAPI. Slides were covered with mounting medium (Vector Laboratories, Burlingame, CA, USA).

Primary Abs used for these studies were specific for: myelin basic protein (MBP, Abcam), CD45 (Abcam), A2B5 (Abcam), glial fibrillary acid protein (GFAP, Abcam), A2B5 (Millipore), adenomatous polyposis coli/CC1 (Millipore), and IBA1 (Abcam). FluoroMyelin staining was order from Invitrogen. Appropriate fluorescent secondary Abs were used (Alexa Fluor, Invitrogen).

## Image analysis

Images were captured by fluorescent microscopy (Nikon Eclipse E600; Nikon, Melville, NY) or confocal microscopy (Zeiss LSM 510; Carl Zeiss, Thornwood, NY). Approximately 8–10 images were captured per slice to cover most of the total area of the slice (excluding the edges), thus removing any bias or variations in image acquisition. Five slices were quantified per treatment/control and the experiment was repeated three times using cultures from different mice. Image acquisition settings were kept the same across different treatments. Myelinated axons were quantified by confocal microscopy as described previously (*Huang et al., 2011*). The intencity of MBP and FluoroMyelin immunostaining, and cell numbers of CD45, IBA1, or GFAP per field was determined using ImagePro software (Media Cybernetics). The numbers of CD45, IBA1, or GFAP-positive cells were counted in a blinded fashion either from representative ×20 or × 40 objective images or a series of images derived from Z-stack imaging.

## Preparation of infiltrating MNCs from the CNS

To acquire CNS MNCs, spinal cords and brains were mechanically dissociated and incubated with Liberase (Roche, Nutley, NJ) for 30 min, passed through a 70 μm cell strainer and washed with cold PBS. Cells were then fractionated on a 70/30% Percoll (Sigma-Aldrich) gradient by centrifugation at 2000 rpm for 20 min and MNCs were collected from the interface and washed with PBS.

## Cytokine measurement by ELISA

Spleen was mechanically dissociated through a 70 μm cell strainer (Falcon, Tewksbury, MA) and incubated with red blood cell lysis buffer (Miltenyi) ~2 min. Harvested cells were washed with cold PBS before *in vitro* stimulation. Splenocytes at $1.0 \times 10^6$ cells/ml were cultured in triplicates in RPMI 1640 supplemented with 10% FBS in 24-well plates and stimulated with 25 μg/ml $MOG_{35-55}$ for 72 h. Supernatants were collected and assayed for IL-6 and TNF-α by ELISA Kits (R&D Systems, Minneapolis, MN).

## Flow cytometry

For surface-marker staining, cells were incubated with fluorochrome-conjugated Abs to CD45, CD4, CD8, CD11b, CD11c, CD80, CD86, and MHC II (BD Biosciences, San Jose, CA) at the recommended dilution or isotype control Abs for 30 min on ice. To analyze MOG-specific Th cells, CNS-infiltrating MNCs were stimulated with 25 μg/ml MOG peptide overnight, followed by stimulation with 50 ng/ml PMA and 500 ng/ml ionomycin in the presence of GolgiPlug for 4 h. Cells were surface-stained with mAbs against CD4 and CD8. Cells were then washed, fixed, and permeabilized with Fix & Perm Medium (Invitrogen), and intracellular cytokines were stained with Abs against IL-17, or IFN-γ, IL-10 (BD Biosciences). Foxp3 staining was carried out using a commercial kit, according to the manufacturer's instructions (eBioscience, San Diego, CA). Flow cytometry analysis was performed on FACSAria (BD Biosciences, San Jose, CA) and data were analyzed with FlowJo software (Treestar, Ashland, OR).

## Isolation of primary microglia, astrocytes, and OPCs

Primary microglia, astrocytes, and OPCs were isolated from brain of newborn mouse (P3), by dissociation with Neural Tissue Dissociation Kit (Miltenyi Biotech Inc) and purification with either anti-CD11b, anti-ACSA-2, or anti-CD140a microbeads (Mitenyi Biotech Inc). Microglia were cultured in DMEM/F12 + 10% FBS, 5% HS, 2 mM Penicillin-Streptomycin, and 5 ng/mL M-CSF. Microglia were then either left unstimulated, stimulated with lipopolysaccharide (LPS) (100 ng/mL), or with 100 μg/mL PM. Cells were analyzed using flow cytometry, Q-PCR, western blotting, ELISA, and co-culture assay. To check the morphology of activated microglia, the cells were labeled with IBA1 and counterstained with DAPI. Astrocytes were cultured in DMEM +1% FBS, 1% N2, 2 mM GlutaMax. Astrocytes were then either left unstimulated, stimulated with LPS (100 ng/mL), or with 100 μg/mL PM. Gene expression was analyzed using Q-PCR. OPCs were cultured in DMEM/F12 supplemented with 2% B27, 1 % N2, and 50% B104 cell supernatant. For differentiation, the proliferation medium of OPCs was replaced by DMEM/F12 with 2% B27, 1 % N2, and changed every 2 days. OPCs were stimulated with 2% B27, 1 % N2, 30 ng/mL T3, 50 ng/mL Shh and 50 ng/mL Noggin, with or without 10 ng/mL CNTF as positive controls to induce OLGs. OPCs were then either left unstimulated, stimulated with LPS (100 ng/mL), or with (25, 50, 100 μg/mL) PM for 3 days. Immunostaining for PDGFRα, an OPC marker, and CNPase, a mature OLG marker, was performed 3 days after treatment.

## CD4+ T cells isolation and culture

Spleens were collected from 6 to 8 week-old C57BL/6 female mice to obtain single-cell mixes. Naive CD4+ T cells were cultured for 72 h with 0.5 µg/mL anti-CD3 (Bioxcell) and 1 µg/mL anti-CD28 (Bioxcell). Differentiation into Th17 cells was induced by adding 10 µg/mL anti-IFN-γ (Bioxcell), 2 ng/mL TGF-β (Peprotech), 20 ng/mL IL-6 (Peprotech), and 10 ng/mL IL-1β (Peprotech). Differentiation into Th1 cells was induced by adding 10 µg/mL IL-12 (Bioxcell). CD4+ T cells were then either left unstimulated, stimulated with LPS (100 ng/mL), or with PM (400 µg/mL).

## Western blot analysis

Cells cultured under different treatments were washed with PBS and lysed by cell lysis buffer (Cell Signaling, Danvers, MA) supplemented with 1 mM phenylmethylsulfonyl fluoride and 1× proteinase inhibitor cocktail (Sigma, St. Louis, MO). Protein concentrations of all samples were determined using the Pierce BCA Protein Assay Kit (Thermo, Rockford, IL). Protein samples (equal amount/lane) were separated by 12% SDS-PAGE and transferred onto nitrocellulose membrane. The transformed membrane was blocked for 2 hr followed by incubation with primary antibodies at 4°C overnight. The membrane was washed three times with TBST buffer (50 mM Tris·HCl, pH 7.4, 150 mM NaCl, 0.1% Tween 20) for 5 min each and then incubated with 1:200 diluted anti-rabbit or mouse IgG-horseradish peroxidise (HRP) (Thermo Scientific, Rockford, IL) at room temperature for 1 h. The protein band was detected using Super Signal West Pico Chemiluminescent Substrate (Thermo Scientific, Rockford, IL).

## Q-PCR

Total RNA was extracted from spinal cords using RNeasy Plus Mini Kit (QIAGEN, Valencia, CA) according to the manufacturer's instructions. Reverse transcription was conducted using QuantiTect Reverse Transcription Kit (Qiagen,). Real-time PCR was performed using the Custom RT² Profiler PCR Array according to the manufacturer's instructions (Qiagen), and detection was performed using the ABI Prism 7,500 Sequence Detection System (Applied Biosystems, Foster City, CA). All data were normalized to an average of five housekeeping genes Gusb, Hprt, Hsp90ab1, Gapdh and Actb. Qiagen's online web analysis tool was utilized and gene relative expression was calculated by log2 of −ΔΔCt values from triplicate of PCR. More than two fold changes (log2 < −one or log2 > 1) were considered significant between groups.

## RNA-Seq and data analysis

RNA from PBS- or PM-treated microglia were prepared using RNAprep pure Cell / Bacteria Kit (Cat. #dp430). RNA degradation and contamination was monitored on 1% agarose gels. RNA purity was checked using the NanoPhotometer spectrophotometer (IMPLEN, CA, USA). RNA concentration was measured using Qubit RNA Assay Kit in Qubit 2.0 Flurometer (Life Technologies, CA, USA). RNA integrity was assessed using the RNA Nano 6,000 Assay Kit of the Bioanalyzer 2,100 system (Agilent Technologies, CA, USA). A total amount of 3 µg RNA per sample was used as input material for the RNA sample preparations. Sequencing libraries were generated using NEBNext UltraTM RNA Library Prep Kit for Illumina (NEB, USA) following manufacturer's recommendations and index codes were added to attribute sequences to each sample. The clustering of the index-coded samples was performed on a cBot Cluster Generation System using TruSeq PE Cluster Kit v3-cBot-HS (Illumia) according to the manufacturer's instructions. After cluster generation, the library preparations were sequenced on an Illumina Hiseq platform and 150 bp paired-end reads were generated. Raw data (raw reads) of fastq format were firstly processed through in-house perl scripts. In this step, clean data (clean reads) were obtained by removing reads containing adapter, reads containing ploy-N and low quality reads from raw data. At the same time, Q20, Q30 and GC content the clean data were calculated. All the downstream analyses were based on the clean data with high quality. STAR is used to align clean reads to reference genome. STAR outperforms other aligners by a factor of >50 in mapping speed, while at the same time improving alignment sensitivity and precision. In addition to unbiased de novo detection of canonical junctions, STAR can discover non-canonical splices and chimeric (fusion) transcripts, and is also capable of mapping full-length RNA sequences. HTSeq v0.6.0 was used to count the reads numbers mapped to each gene. And then FPKM of each gene was calculated based on the length of the gene and reads count mapped to this gene. FPKM, expected number of Fragments Per Kilobase of transcript sequence per Millions base pairs sequenced, considers the

effect of sequencing depth and gene length for the reads count at the same time, and is currently the most commonly used method for estimating gene expression levels. We applied DESeq2 algorithm to filter the differentially expressed genes, and DEGs were defined as genes with FDR less than 0.001 and fold change larger than 2. GO Analysis: Gene ontology (GO) analysis was performed to facilitate elucidating the biological implications of unique genes in the significant or representative profiles of the differentially expressed gene in the experiment. We downloaded the GO annotations from NCBI (http://www.ncbi.nlm.nih.gov/), UniProt (http://www.uniprot.org/) and the Gene Ontology (http://www.geneontology.org/). Fisher's exact test was applied to identify the significant GO categories and FDR was used to correct the p-values. Pathway analysis was used to find out the significant pathway of the differential genes according to KEGG database. We turn to the Fisher's exact test to select the significant pathway, and the threshold of significance was defined by p-value and FDR.

## ChIP-Seq and data analysis

Approximately 1.2 million microglia were exposed to the indicated treatments followed by cell preparation according to the SimpleChIP Enzymatic Chromatin IP Kit (Cell Signaling Technology, #9003). Samples were cross-linked for 10 min at room temperature with 1% formaldehyde solution, followed by 5 min of quenching with 125 mM glycine. Then washed twice with cold PBS, and the supernatant was aspirated. Nuclei were fragmented with a Misonix Sonicator 3,000. Sonicated lysates were cleared once by centrifugation and incubated overnight at 4 °C with magnetic beads bound with NF-kB antibody (Cell Signaling Technology, Cat. No. 8242 s) to enrich for DNA fragments. For the preparation of the magnetic beads bound with NF-kB antibody, 70 μl of Protein G Dynabeads (Life Technologies) was blocked with 0.5% (w/v) BSA in PBS first, and then magnetic beads were bound with 10 μg anti-NFkB. After overnight incubation with the cleared sonicated lysates, magnetic beads were washed with RIPA buffer, 1 M NH4HCO3. DNA was eluted in elution buffer. Cross-links were reversed overnight. Protein was digested using Proteinase K, and DNA was purified with HiPure Gel Pure DNA Mini Kit. Purified ChIP DNA was used to prepare Illumina multiplexed sequencing libraries. Libraries were prepared following the NEB/NEBNext Library Quant Kit for Illumina (E7630S). Amplified libraries were size-selected using a 2% gel to capture fragments between 200 and 500 bp. Libraries were quantified by Agilent 2,100. Libraries were sequenced on the Illumina NovaSeq6000.

For data analysis, Quality distribution plot and base content distribution were generated by FASTQC. Before read mapping, clean reads were obtained from the raw reads by removing the adaptor sequences. Paired-end ChiP-Seq reads were aligned using BWA mem (v.0.7.8) against the GRCm39/mm10 mouse genome assembly with default settings. PCR duplicates were not present in the dataset. Alignments were fifiltered with SAMtools (v1.3) to exclude reads with mapping quality <30, not properly paired, aligned to mitochondrial genome, and aligned to ENCODE blacklist regions (ENCODE Project Consortium, 2012). For peak calling, MACS2 callpeak (v2.1.1) were called on individual replicates for each ChIP (treatment) and Input (control) pair, using q value < 0.05. The HOMER's findMotifsGenome.pl tool was used for Motif analysis. Peaks were annotated by the function of annotatePeak of ChIPseeker. Reads distributions (from bigwig) across gene are presented as an average plot (average of reads signals across the targeted genes). The deeptools tool is used for this analysis. Differential peaks were then selected with the absolute value of the log2 fold change was one at a p-value < 0.05 using DESeq2.

## Statistical analysis

Statistical analyses were performed using GraphPad Prism six software (GraphPad, La Jolla, CA). Data are presented as mean ± SD. When comparing multiple groups, data were analyzed by analysis of variance (ANOVA) with Tukey's multiple comparisons test. A significance criterion of $p < 0.05$ was used for all statistical analysis.

## Acknowledgements

This study was supported by the Chinese National Natural Science Foundation (Grant No. 82071396, 31970771, 41771220, 81801195, U1804178), the Natural Science Foundation of Shaanxi Province, China (Grant No. 2019KJXX-022), the Fundamental Research Funds for the Central Universities (Grant No. GK202105002, GK202007022, 2020CSLZ010, S202110718278, 2019TS080).

## Additional information

### Funding

| Funder | Grant reference number | Author |
|---|---|---|
| National Natural Science Foundation of China | 82071396 | Yuan Zhang |

The funders had no role in study design, data collection and interpretation, or the decision to submit the work for publication.

### Author contributions

Bing Han, Investigation, Methodology, Writing – review and editing; Xing Li, Yuan Zhang, Conceptualization, Data curation, Formal analysis, Funding acquisition, Investigation, Methodology, Project administration, Supervision, Writing - original draft; Ruo-Song Ai, Xin Deng, Wen Ma, Yan Zhang, Yan Xu, Writing – review and editing; Si-Ying Deng, Ze-Qing Ye, Investigation; Shun Xiao, Jing-Zhi Wang, Li-Mei Wang, Chong Xie, Supervision, Writing – review and editing

### Author ORCIDs

Bing Han http://orcid.org/0000-0002-7301-4126
Yuan Zhang http://orcid.org/0000-0002-2463-4599

### Ethics

All experimental procedures and protocols of mice were approved by the Committee on the Ethics of Animal Experiments of Shaanxi Normal University (No. ECES-2015-0247) and were carried out in accordance with the approved institutional guidelines and regulations. C57BL/6 mice (8-10 weeks of age) were purchased from the Fourth Military University (Xi'an, China).

### Decision letter and Author response

Decision letter https://doi.org/10.7554/eLife.72247.sa1
Author response https://doi.org/10.7554/eLife.72247.sa2

## Additional files

### Supplementary files

- Transparent reporting form
- Source data 1. Data generated or analyzed during this study.

### Data availability

Figure 1 - Source Data 1, Figure 2 - Source Data 1, Figure 3 - Source Data 1, Figure 3 - Source Data 1, Figure 1-figure supplement 1- Source Data 1 and Figure 4-figure supplement 1- Source Data 1 contain the numerical data used to generate the figures. Sequencing data are available through the Zhang Y, Han B, 2021 Atmospheric particulate matter aggravates CNS demyelination via TLR-4/NF-κB-mediated microglia pathogenic activities, https://www.ncbi.nlm.nih.gov/geo/query/acc.cgi?acc=GSE183099, NCBI Gene Expression Omnibus, GSE183099, accession.

The following dataset was generated:

| Author(s) | Year | Dataset title | Dataset URL | Database and Identifier |
|---|---|---|---|---|
| Zhang Y, Han B | 2021 | Atmospheric particulate matter aggravates CNS demyelination via TLR-4/ NF-κ B-mediated microglia pathogenic activities | https://www.ncbi.nlm.nih.gov/geo/query/acc.cgi?acc=GSE183099 | NCBI Gene Expression Omnibus, GSE183099 |

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
