## [Editor Report]

The study by Han et al. uses a rodent model of demyelination to investigate the effects of atmospheric particulate matter (PM) on CNS demyelination. The authors provide evidence that PM can promote and exacerbate demyelination which is associated with increased microglial activation and inflammation in the rodent central nervous system. These findings further our understanding of how environmental factors can influence human diseases.

---

## [Decision Letter]

**Decision letter after peer review:**

Thank you for submitting your article "Atmospheric particulate matter aggravates CNS demyelination via TLR-4/NF- κB-mediated microglia pathogenic activities" for consideration by *eLife*. Your article has been reviewed by 2 peer reviewers, and the evaluation has been overseen by a Reviewing Editor and Tadatsugu Taniguchi as the Senior Editor. The reviewers have opted to remain anonymous.

The authors investigated PM on myelination in the CNS using a well-established animal model of demyelination. They showed that the PM-induced demyelination causes neurological dysfunction that is associated with microglial activation and markers of inflammation. They also show that PM can activate the TLR/NFkB signaling using in vitro model of microglial activation. They use the in vitro data to claim in vivo effects are also due to TLR4/NKFB activation by PM, however, additional evidence is needed to support this claim. Overall, the authors provide evidence that PM can promote and exacerbate demyelination which is associated with increased inflammation in the rodent central nervous system.

Essential revisions:

1) The title and some claims should be revised given that PM effects on demyelination in vivo do not provide strong evidence for TLR4/NFKB role. The in vitro microglia data indicate activation of TLR4/NFKB by PM but this must be experimentally validated in vivo or the claims adjusted to reflect that the effects of PM in vivo may be in part mediated by activation of TLR4/NFKB. Ultimately, the mechanism of PM exacerbation of demyelination is not elucidated in vivo (i.e. no pharmacological or genetic manipulation of TRL4/NFKB) and may or may not involve the TLR4/NFKB pathway. It is likely they are activated but whether they are primary drivers of the demyelination as the title implies is not demonstrated here. The authors should alter the claims to better reflect the data and its implications or add additional experiments to support the in vivo role of TLR4/NFKB.

2) In Figure 1, the authors showed that for both central and peripheral immunity, the percentages of Th17 (CD4^+^ IL17+) and Th1 (CD4^+^ IFN-γ+) cells were significantly increased under PM exposure. PM has already been reported to have some effects on peripheral immunity. The authors remain to examine the effects of PM on different T cells subsets in vitro? Additionally, it is not clear that in Figure 1G, the % of CD45+ cells reached to 130 +/- 20%, is this correct?

3) In both Figure 2 and Figure 3, the authors show the activation of microglia and astrocytes by PM. Although in Figure 1, the authors mentioned that activated IBA1+ microglia and A2B5+ OPCs accumulated significantly in demyelinated injured areas in PM-treated mice, while PM inhalation had no significant effect on GFAP+ astrocytes. Microglia showed activation results by PM in three complementary animal models, which led the authors to select the final follow-up study subjects of microglia. Given the above results, the authors should examine the direct effects of PM on primary astrocytes, at least in vitro.

4) In Figure 2C, the column results in "Demyelination scores" showed reduced demyelination in the PM group, and the 4+2 groups even reduced more. These appear to contradict the imaging graphs and conclusions made in manuscript.

5) For the in vitro modeling, how was the dosage of PM selected? How does the dose compare to what cells would be exposed to in the in vivo model? Also, in Material and Methods, the dose of PM used in microglia culture was 200 mg/ml, but in Figure 4 the dose of PM was 100 μg/mL, why is the different dose used?

6) In Figure 4. The authors proposed that treatment of purified primary OPC with microglia-conditioned medium (MCM) in vitro prevented OPC differentiation. OPC are essential for remyelination after central nervous system injury. Therefore, it is necessary for the authors to investigate the direct effects of PM on OPC and thus better illustrate the effects of PM exposure on CNS demyelinating diseases.

7) CD80" was mentioned in the figure legend of 4B and in the Results section, but it was not shown in the figure. It should be added to the figure, or be removed from the legend and Results.

8) Results and figures of Figure 4E and F are matched each other; however, their legends are opposite to figures and Results. Please check and correct.

9) In Figure 5. The authors predicted targeted genes for PM to induce microglial activation via the TLR-4/NF-κB signaling axis by RNA-seq and ChIP-seq. However, the manuscript would benefit if the authors also discussed the role of predicted target genes in PM-induced microglial activation, and what might be done subsequently.

[Editors' note: further revisions were suggested prior to acceptance, as described below.]

Thank you for resubmitting your work entitled "Atmospheric particulate matter aggravates CNS demyelination in mice via TLR-4/NF-κB-mediated microglia pathogenic activities" for further consideration by *eLife*. Your revised article has been reviewed by 2 peer reviewers and the evaluation has been overseen by Tadatsugu Taniguchi (Senior Editor) and a Reviewing Editor.

The manuscript has been improved but there are some remaining issues that need to be addressed, as outlined below:

Essential revisions:

1) Please revise the title. We commend authors on the added experiments testing inhibitors of TLR4 and NFkB in the model which produced partial effects. As the authors point out, this newly added data suggest that the effects of PM on EAE are "maybe partially, by TLR4/NFkB activation in vivo" but the title makes it seem like the effects are mediated entirely by TLR4/NFkB on microglia. The data do not support the title as written. Consider altering title to something like the following: "Atmospheric particulate matter aggravates CNS demyelination through involvement of TLR-4/NF-κB signaling and microglial activation." The study supports 1) PM worsens demyelination, 2) TLR4/NFkB are involved but not the only mediators, 3) microglial activation occurs although the causative link to aggravating demyelination is not firmly established. Overall, the title should be reflective of the finding and not overstate data.

2) The authors provide data that astrocytes are not responsive to PM. Additional discussion on selective response of microglia to PM would be helpful.

3) The EM images should be clearly labeled with arrow heads for the demyelinated axons in PM groups (Figure 2D and 3D).

4) Please provide details of the "G-ratio" in figure legends.

---

## [Author Response]

Essential revisions:1) The title and some claims should be revised given that PM effects on demyelination in vivo do not provide strong evidence for TLR4/NFKB role. The in vitro microglia data indicate activation of TLR4/NFKB by PM but this must be experimentally validated in vivo or the claims adjusted to reflect that the effects of PM in vivo may be in part mediated by activation of TLR4/NFKB. Ultimately, the mechanism of PM exacerbation of demyelination is not elucidated in vivo (i.e. no pharmacological or genetic manipulation of TRL4/NFKB) and may or may not involve the TLR4/NFKB pathway. It is likely they are activated but whether they are primary drivers of the demyelination as the title implies is not demonstrated here. The authors should alter the claims to better reflect the data and its implications or add additional experiments to support the in vivo role of TLR4/NFKB.

Following this advice, TAK242 and PDTC, pharmacological inhibitors of TLR4 and NF-κB, were administrated to the EAE mice with or without PM exposure. Results shown that both of the inhibitors effectively reversed the deterioration of the disease caused by PM exposure in the EAE model (Figure 4J, page 13, line 282-284), demonstrating that the effect of PM was mediated, maybe partially, by TLR4/ NF-κB activation in vivo.

2) In Figure 1, the authors showed that for both central and peripheral immunity, the percentages of Th17 (CD4^+^ IL17+) and Th1 (CD4^+^ IFN-γ+) cells were significantly increased under PM exposure. PM has already been reported to have some effects on peripheral immunity. The authors remain to examine the effects of PM on different T cells subsets in vitro? Additionally, it is not clear that in Figure 1G, the % of CD45+ cells reached to 130 +/- 20%, is this correct?

We thank the reviewer for this advice. According to your suggestion, we had evaluated the effect of PM on Th17/Th1 cell differentiation. The results showed that PM incubation in vitro promoted Th1 cells, while had no significant effect on Th17 cell differentiation (Figure 4—figure supplement 1A, page 7, line 142-145). We apologized for the statistical error of CD45+ cells, and have been corrected in the revised manuscript (Figure 1G).

3) In both Figure 2 and Figure 3, the authors show the activation of microglia and astrocytes by PM. Although in Figure 1, the authors mentioned that activated IBA1+ microglia and A2B5+ OPCs accumulated significantly in demyelinated injured areas in PM-treated mice, while PM inhalation had no significant effect on GFAP+ astrocytes. Microglia showed activation results by PM in three complementary animal models, which led the authors to select the final follow-up study subjects of microglia. Given the above results, the authors should examine the direct effects of PM on primary astrocytes, at least in vitro.

Following this advice, we examine the direct effects of PM on primary astrocytes in vitro. In contrast to the effect of LPS, PM stimulation did not activate astrocytes in vitro (Figure 4—figure supplement 1B, page 12-13, lines 260-266).

4) In Figure 2C, the column results in "Demyelination scores" showed reduced demyelination in the PM group, and the 4+2 groups even reduced more. These appear to contradict the imaging graphs and conclusions made in manuscript.

We apologize for this mistake. Following this suggestion, we re-examined the statistical analysis of the “demyelination score” and present the data in the revised (Figure 2C).

5) For the in vitro modeling, how was the dosage of PM selected? How does the dose compare to what cells would be exposed to in the in vivo model? Also, in Material and Methods, the dose of PM used in microglia culture was 200 mg/ml, but in Figure 4 the dose of PM was 100 μg/mL, why is the different dose used?

Thanks to the reviewer for raising this question, the doses of PM in the animal model of our study was chosen based on the calculation of population exposure doses and according to the previous studies (Page 21, line 452-460). Because the human body is a complex system, the accurate concentration of PM in the brain is currently unknown. Therefore, in vitro doses cannot be compared to cells exposed in in vivo models. We choose the dosage of PM in the in vitro modeling according to our preliminary experiments of dose optimization. In our study, PM dose used in microglial cell culture is 100 μg/mL, and we apologize for this error, which we have modified in the revised manuscript (Page 29, line 621-622).

6) In Figure 4. The authors proposed that treatment of purified primary OPC with microglia-conditioned medium (MCM) in vitro prevented OPC differentiation. OPC are essential for remyelination after central nervous system injury. Therefore, it is necessary for the authors to investigate the direct effects of PM on OPC and thus better illustrate the effects of PM exposure on CNS demyelinating diseases.

According to the reviewer’s advice, we tested the direct effect of PM on OPC in vitro. The results showed that PM treatment inhibited the OPC differentiation in a dose-dependent manner (Figure 4—figure supplement 1C-E, page 13, line 266-271).

7) CD80" was mentioned in the figure legend of 4B and in the Results section, but it was not shown in the figure. It should be added to the figure, or be removed from the legend and Results.

We apologize for this mistake and had corrected it in the revised manuscript (Page 11, line 237-238; Figure legend of 4B).

8) Results and figures of Figure 4E and F are matched to each other; however, their legends are opposite to figures and Results. Please check and correct.

We apologize for the confusion. This error was corrected in the revised manuscript (Page 42-43, line 987-996).

9) In Figure 5. The authors predicted targeted genes for PM to induce microglial activation via the TLR-4/NF-κB signaling axis by RNA-seq and ChIP-seq. However, the manuscript would benefit if the authors also discussed the role of predicted target genes in PM-induced microglial activation, and what might be done subsequently.

We are grateful for the reviewer’s advice. According to the constructive suggestion, we discussed the role of predicted target genes in PM-induced microglial activation, and added our subsequent plan in the revised manuscript (Page 19-20, line 409-422).

[Editors' note: further revisions were suggested prior to acceptance, as described below.]

Essential revisions:1) Please revise the title. We commend authors on the added experiments testing inhibitors of TLR4 and NFkB in the model which produced partial effects. As the authors point out, this newly added data suggest that the effects of PM on EAE are "maybe partially, by TLR4/NFkB activation in vivo" but the title makes it seem like the effects are mediated entirely by TLR4/NFkB on microglia. The data do not support the title as written. Consider altering title to something like the following: "Atmospheric particulate matter aggravates CNS demyelination through involvement of TLR-4/NF-κB signaling and microglial activation." The study supports 1) PM worsens demyelination, 2) TLR4/NFkB are involved but not the only mediators, 3) microglial activation occurs although the causative link to aggravating demyelination is not firmly established. Overall, the title should be reflective of the finding and not overstate data.

Following your constructive suggestion, the title was altered as:

“Atmospheric particulate matter aggravates CNS demyelination through involvement of TLR-4/NF-κB signaling and microglial activation”.

2) The authors provide data that astrocytes are not responsive to PM. Additional discussion on selective response of microglia to PM would be helpful.

We thank the reviewer for this advice. According to your suggestion, we added a discussion about the selective response of microglia to PM (Page 17, line 361-365).

3) The EM images should be clearly labeled with arrow heads for the demyelinated axons in PM groups (Figure 2D and 3D).

According to this advice, we labeled demyelinated axons with arrows heads on the EM images of the PM group (Figure 2D and 3D).

4) Please provide details of the "G-ratio" in figure legends.

Following this advice, we provided the details of the "G-ratio" in the figure legend (Page 41, line 953-954; Page 42, line 972-973).